# Gastric bypass prevents diabetes in genetically modified mice and chemically induced diabetic mice

Chenyu Zhu[1]☯, Rui Xu[1]☯, Yuxin Li[2], Michael Andrade[2], Deng Ping Yin[2]*

1 The First College of Clinical Medical Science, CTGU, and Yichang Central People's Hospital, Yichang, Hubei, China, 2 The Department of Surgery at University of Chicago, Chicago, Illinois United States of America

☯ These authors contributed equally to this work.
* dyin@surgery.bsd.uchicago.edu

## Abstract

Obese subjects have increase probabilities of developing type 2 diabetes (T2D). In this study, we sought to determine whether gastric bypass prevents the progression of prediabetes to overt diabetes in genetically modified mice and chemically induced diabetic mice. Roux-en-Y gastric bypass (RYGB) was performed in C57BL/KsJ-db/db null (BKS-db/db,) mice, high-fat diet (HFD)-fed NONcNZO10/LtJ (NZO) mice, C57BL/6 db/db null (B6-db/db) mice and streptozotocin (STZ)-induced diabetic mice. Food consumption, body weight, fat mass, fast blood glucose level, circulating insulin and adiponectin and glucose tolerance test were analyzed. The liver and pancreatic tissues were subjected to H&E and immunohistochemistry staining and islet cells to flow cytometry for apoptotic analysis. RYGB resulted in sustained normoglycemia and improved glucose tolerance in young prediabetic BKS-db/db mice (at the age of 6 weeks with hyperglycemia and normal insulinemia) and HFD-fed NZO and B6-db/db mice. Remarkably, RYGB improved liver steatosis, preserved the pancreatic β-cells and reduced β-cell apoptosis with increases in circulating insulin and adiponectin in young prediabetic BKS-db/db mice. However, RYGB neither reversed hyperglycemia in adult diabetic BKS-db/db mice (12 weeks old) nor attenuated hyperglycemia in STZ-induced diabetic mice. These results demonstrate that gastric bypass improves hyperglycemia in genetically modified prediabetic mice; however, it should be performed prior to β-cells exhaustion.

## Introduction

Obesity is associated with insulin resistance and type 2 diabetes (T2D) and is considered a public health crisis, responsible for an exponential increase in health care costs [1]. In response to insulin resistance, pancreatic β-cells compensate by increasing insulin secretion characterized by hyperglycemia and hyperinsulinemia or normal insulinemia [2]. This metabolic setting can be maintained for long periods of time [3–6]. Eventually, in some individuals who are genetically predisposed to develop T2D, pancreatic β-cells are unable to maintain the increased

**Data Availability Statement:** All relevant data are within the manuscript and its Supporting information files.

**Funding:** NIHDDK DK020595 to the Metabolic Core of the University of Chicago NIDDK P30 DK42086

to the Tissue and Cell Models Core of the
University of Chicago DDRCC.

**Competing interests:** The authors have declared
that no competing interests exist.

demand for insulin, and overt diabetes ensues [7–9]. To prevent or delay the development of overt diabetes, bariatric surgery, including gastric bypass, have been initiated. However, while gastric bypass effectively improves obesity and insulin resistance, complete resolution of diabetes occurs in only 42.3 to 59.4% of patients [10–12]. A shorter duration of diabetes prior to surgery corresponds to a higher rate of diabetes remission. Likewise, less favorable outcomes may be associated with longer duration of diabetes, poor preoperative hyperglycemia control and reduced β-cell function [13,14]. Thus, an early surgical intervention like gastric bypass in obese patients may be a viable strategy to prevent the progression of diabetes.

Frequently used animal models of obesity and diabetes are rodents that are exposed to obesogenic diets predisposing the animals to gaining excessive weight [15]. An established mouse RYGB model induces significant weight loss, reduces body fat, improves insulin sensitivity and ameliorates liver steatosis in high-fat diet (HFD)-induced obese mice [16–19]. However, HFD-induced obesity and insulin resistance reveal hyperinsulinemia, insulin resistant and hyperglycemia without β-cell failure or insulin depletion [20]. Mice with mutations of individual or several genes are susceptible to develop obesity and diabetes. In leptin receptor deficient db/db mice, C57BL/KsJ-db/db nude (BKS-db/db) mice carry a mutation in the leptin receptor gene characterized by extreme obesity and early onset of hyperglycemia and hyperinsulinemia or normal insulinemia. An uncontrolled rise in blood sugar or overt diabetes develops due to severe depletion of pancreatic β-cells with aging [21]. Young prediabetic BKS-db/db mice ($\leq$ 6 weeks old) show early metabolic derangements and hyperglycemia and normal insulinemia before the onset of β-cell failure [22]; thereby, providing an opportunity to test whether gastric bypass can prevent the progression of prediabetes to overt diabetes. C57BL/6 background leptin receptor deficient (B6-db/db) homozygous mice reveal compensatory hyperplasia of β-cells and continued hyperglycemia without depletion of β-cells, which are used to model T2D from 1 to 3 phases [23,24]. This allows us to test whether gastric bypass improves obesity and hyperglycemia in prediabetic states, compared to BKS-db/db mice. T2D results from a combination of genetic and environmental factors and is increasingly attributed to environmental factors, particularly HFD. NONcNZO10/LtJ (NZO) is a recombinant congenic strain developed at the Jackson Laboratory to model human obesity induced T2D. These mice develop overt diabetes in 40% of male mice, and HFD accelerates the progression to fasting hyperglycemia in 100% of male mice [25,26]. This model provides us the opportunity to test whether gastric bypass can prevent diabetes induced by genetical and environmental factors. In the chemically induced diabetic model, pancreatic β-cells are severely damaged by administration of streptozotocin (STZ), mimicking type 1 diabetes (T1D) phenotype [27,28]. The availability of mouse insulin promoter (MIP)-driven luciferase transgenic (*MIP-luc*) mice provides an opportunity to visualize insulin promoter activity, which lets us determine whether gastric bypass can improve β-cell activity when β-cells are exhausted.

In this study, we ask whether gastric bypass can prevent the development of prediabetes to overt diabetes, using genetically modified prediabetic and diabetic mice and chemically induced diabetic *MIP-luc* mice. Our results show the efficacy of gastric bypass in preventing diabetes if it is performed prior to severe β-cell exhaustion.

## Materials and methods

### Mice and surgery

C57BL/KsJ-db/db nude (BKS-db/db), C57BL/6-db/db nude (B6-db/db) and NONcNZO10/LtJ (NZO) mice were purchased from the Jackson Laboratory (Bar Harbor, ME). Mouse insulin promoter (MIP)-driven luciferase transgenic (*MIP-luc*) mice were used to analyze insulin promoter activity [29]. Mice were housed at 23˚C on a 07:00–19:00 light cycle. Young BKS-db/

**Table 1. Mouse models and diet.**

| Mice | Surgery Age[*] | Diet[**] | Surgery |
|---|---|---|---|
| BKS-db/db | 6w | Chow | RYGB or Sham-PF[***] |
| NZO | 6w | HFD | RYGB or Sham-PF |
| STZ | 10w | Chow | RYGB or Sham-PF |
| B6-db/db | 6w | Chow | RYGB or Sham-PF |

[*]: BKS-db/db, B6-db/db and NZO mice underwent RYGB or sham surgery at 6 weeks of age, and streptozotocin (STZ)-induced diabetic mice underwent RYGB or sham surgery at 10 weeks of age.

[**]: BKS-db/db, B6-db/db and STZ mice were fed with chow for whole study period. NZO mice were fed with chow up to 6 weeks of age and underwent RYGB or sham surgery followed by high-fat diet (HFD, 60% fat, Bio-Serv) feeding up to the endpoint.

[***]: Sham-PF, mice undergo sham surgery with pair feeding. The amount of diet provided to the mouse with sham surgery is matched to that consumed by the mouse with RYGB.

db mice (6 weeks old with blood glucoses of 300–350mg/dl and normal insulinemia), adult BKS-db/db mice ($\geq$ 12 weeks old with blood glucoses of $\geq$ 400 mg/dl and reduced plasma insulin levels) and adult B6-db/db mice ($\geq$ 12 weeks old with blood glucoses of $\leq$ 300 mg/dl) were used. All BKS-db/db and B6-db/db mice were fed chow for the duration of the experiment. NZO mice were fed with chow up to 6 weeks of age, after which gastric bypass and sham surgeries were performed, followed by a HFD until the end of the experiment. To test whether RYGB reverses chemically induced diabetes, *MIP-luc* mice were made diabetic (blood glucoses were $\geq$ 300 mg/dl for more than two consecutive days) by a single intravenous injection of STZ (170 mg/kg, Sigma-Aldrich, St. Louis, MO). The mouse models and diet feeding are described in Table 1 and the S1 File.

RYGB was performed with a success rate of $\geq$80% as described in our previous reports and the S1 File [16,17,30], and RYGB with a small gastric pouch was used in this study. Blood glucose was tested before and after surgery using a glucose meter (SureStep, LifeScan, Inc., Milpitas, CA). In the sham procedure, the stomach, duodenum and jejunum were mobilized, and the esophagus, stomach and jejunum were clamped without incision followed by the closure of the abdomen. This study was carried out in strict accordance with the recommendations in the Guide for the Care and Use of Laboratory Animals of the National Institutes of Health. At the endpoint (12 weeks post-surgery), mice with RYGB or sham surgery were sacrificed under the anesthesia of isoflurane, and the livers were collected for H&E staining. All surgery was performed under isoflurane, and all efforts were made to minimize suffering. The protocol (ACUP number: 72357) was approved by the University of Chicago Institutional Animal Care and Use Committee.

## Whole body composition

Body mass was measured using the mq10 NMR analyzer (Bruker Optics Inc., Billerica, MA) following a 2-hr fast.

## Intraperitoneal glucose tolerance tests (IPGTT)

Mice were fasted for 4 hrs prior to the IPGTT. Blood was sampled from the tail vein at 0, 30, 60, 90 and 120 min after an intraperitoneal injection of 20% dextrose dosed at 2g/kg body weight. Blood glucose was measured using a blood glucose meter (SureStep, Lifescan, Inc.).

The area under the cure (AUC) was calculated using the trapezoidal rule [31]. Any result exceeding the maximum reading of meter (showing HIGH) was recorded as 600 mg/dL.

## Plasma insulin and adiponectin analyses

Circulating insulin and adiponectin were analyzed by the MilliPlex map kit using the Luminex 200 System analyzer (Millipore, Co., Billerica, MA) following company's instruction.

## Flow cytometry for annexin V positive cell analysis

Single islet cells were collected by collagenase digestion and Ficoll gradient procedure [28,32] at 4 weeks post-surgery. Cells were subjected to fluorescein-labeled annexin V (Annexin V-FITC) in concert with propidium iodide (PI) using the ApoScreen Annexin V Apoptosis Kit (SouthernBiotech, Birmingham, Alabama). Annexin V-positive cells were analyzed by flow cytometry (3-laser BD LSRII system, San Jose, CA). Cells in early apoptosis are Annexin V-FITC-positive and PI-negative, and cells in late apoptosis are both FITC Annexin V- and PI-positive.

## Bioluminescence imaging (BLI)

BLI in *MIP-luc* mice allows for direct visualization of β-cell viability under isoflurane anesthesia [29]. Luciferin (Roche Diagnostics, Indianapolis, IN) was injected intravenously at a dose of 50mg/kg. Mice were housed inside a light-tight box and imaged with an ICCD camera (Hamamatsu C2400-32, Hopkinton, MA). Light emission through the ventral body was detected as photon counts over a standardized area of the organs of interest using the ARGUS-50 software for image processing [29,33].

## Statistics

Statistical significance was analyzed by ANOVA test (StatView 4.5, Abacus Concepts, Berkeley, CA). *P*-value of $< 0.05$ was considered significant. Results were presented as the mean ± SEM. The level of significance was set at a probability of $p < 0.05$.

# Results

## Gastric bypass improves hyperglycemia and glucose tolerance in young prediabetic BKS-db/db mice

RYGB or sham surgery was performed in young prediabetic BKS-db/db mice showing hyperglycemia and normal insulinemia. Diet intake was analyzed in untreated (BKS-db/db Control), sham surgery treated (BKS-db/db-SHAM) and RYGB-treated (BKS-db/db-RYGB) mice. Fig 1a shows that chow consumption in BKS-db/db-Control mice was 6.2 ± 0.5 g/day at 6 weeks of age and increased to 7.2 ± 0.5 g/day at 18 weeks of age. Chow intake in BKS-db/db-RYGB mice was reduced immediately post-surgery, and BKS-db/db-SHAM mice revealed the similar change to RYGB mice in the first 2 weeks post-surgery. However, diet consumption was increased 4 weeks after surgery in BKS-db/db-SHAM mice and reduced through the end point (12 weeks post-surgery) in BKS-db/db-RYGB mice. To test the effects of caloric restriction and surgical stress on the induction of weight loss and fat reduction, a separate group of mice underwent sham surgery and were pair fed (SHAM-PF). The amount of diet provided to SHAM-PF mice is matched to that consumed by the mice with RYGB. The results in Fig 1b show that RYGB decreased body weight, and SHAM-PF experienced similar weight changes in BKS-db/db mice (BKS-db/db-SHAM-PF vs. BKS-db/db-RYGB, $p > 0.05$, Fig 1b). Fat mass in BKS-db/db mice was enhanced compared to wild-type (WT) C57BL/6 LEAN mice and

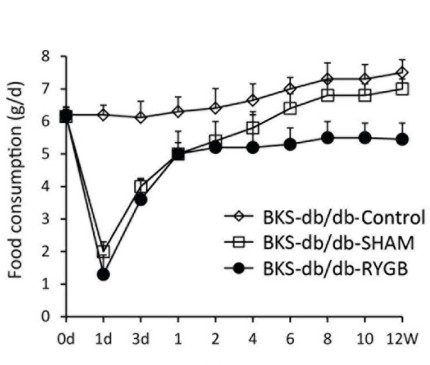

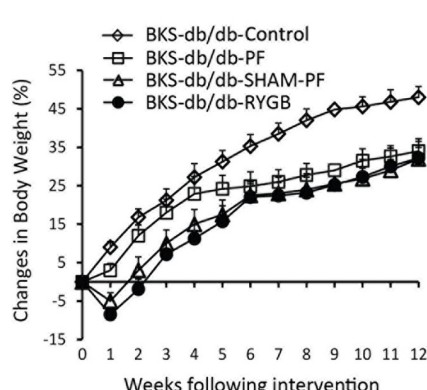

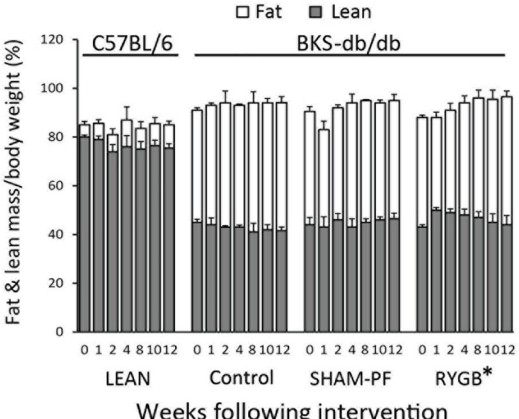

**Fig 1. Diet consumption, body weight and body composition in BKS-db/db mice.** a. Diet consumption. Chow consumption was calculated starting at 6 weeks of age for 12 weeks (mice at 6 weeks of age was defined as 0 week). BKS-db/db-Control, untreated BKS-db/db mice (n = 5); BKS-db/db-SHAM, BKS-db/db mice with sham surgery (n = 4), and BKS-db/db-RYGB, BKS-db/db mice with RYGB (n = 5). b. Weight loss and gain. BKS-db/db-Control, untreated BKS-db/db mice (n = 8); BKS-db/db-PF, pair-fed BKS-db/db mice (n = 5); BKS-db/db-SHAM-PF, BKS-db/db mice underwent sham surgery and were pair-fed (n = 6), and BKS-db/db-RYGB, BKS-db/db mice with RYGB (n = 8). c. Fat and lean mass changes. Body composition was measured by a NMR analyzer, as described in the Methods and our previous report [16]. Fat and lean mass were expressed as fat or lean mass percentage of body weight. LEAN, chow fed C57BL/6 mice (n = 6); Control, untreated BKS-db/db mice (n = 8); SHAM-PF, BKS-db/db mice with sham surgery and pair-feeding (n = 6), and RYGB, BKS-db/db mice with RYGB (n = 8). *: BKS-db/db RYGB vs. BKS-db/db Control, $p < 0.05$, and BKS-db/db RYGB vs. BKS-db/db SHAM-PF, $p = 0.1$, from the first week to 12 weeks post-surgery.

significantly reduced by RYGB (BKS-db/db-RYGB vs. BKS-db/db-Control, $p < 0.05$ from the 1st week to 12 weeks post-surgery, Fig 1c).

Blood glucose was increased in young BKS-db/db mice ($\geq$ 300mg/dL), and RYGB resulted in a persistent normoglycemia, whereas blood glucose in SHAM-PF mice had only transient declines and increased 2 weeks post-surgery (Fig 2a). SHAM-PF ($p < 0.01$) improved glucose tolerance compared to control BKS-db/db mice in the first two weeks after surgery (Fig 2b),

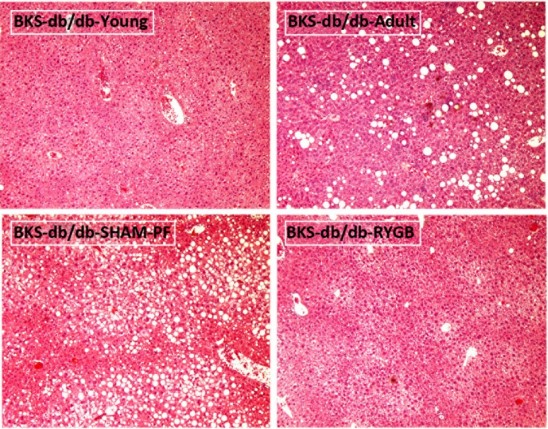

**Fig 2. RYGB maintains normoglycemia and improves glucose tolerance and liver steatosis when performed in young prediabetic BKS-db/db mice (6 weeks of age).** a. Blood glucose levels. B6-LEAN, Chow-fed WT C57BL/6 (n = 5); BKS-db/db-Control, untreated BKS-db/db mice (n = 5); BKS-db/db-SHAM-PF, BKS-db/db mice with sham surgery and pair-feeding (n = 6), and BKS-db/db-RYGB, BKS-db/db mice with RYGB (n = 8). Mice at 6 weeks of age was defined as 0 week. The shaded area represents a normal range of lean mouse blood glucose levels. BKS-db/db-RYGB vs. BKS-db/db-Control $p < 0.001$ from the $1^{st}$ week to 12 weeks post-surgery. BKS-db/db-SHAM-PF vs. BKS-db/db-Control, $p < 0.01$ at the first two weeks ($p < 0.05$), $p > 0.05$ from 4 to 10 weeks and $p < 0.05$ at 12 weeks post-surgery. b. Intraperitoneal glucose tolerance tests (IPGTT). B6-LEAN, C57BL/6 mice (n = 5); BKS-db/db-Control, untreated BKS-db/db mice (n = 5); BKS-db/db-SHAM-PF (n = 5), and BKS-db/db-RYGB (n = 7) mice. *: BKS-db/db-RYGB vs. BKS-db/db-Control and BKS-db/db-SHAM-PF, $p < 0.001$ and #: BKS-db/db-SHAM-PF vs. BKS-db/db-Control, $p < 0.05$. c. Liver H&E stanning (x 100). BKS-db/db-Young, young prediabetic BKS-db/db (6 weeks of age); BKS-db/db-Adult, adult diabetic BKS-db/db (12 weeks of age); BKS-db/db-SHAM-PF (12 weeks post-surgery) and BKS-db/db-RYGB (12 weeks post-surgery). The figures represent one of three pathological examinations in each group.

whereas this improvement persisted in RYGB mice ($p < 0.001$), but not in SHAM-PF mice ($p > 0.05$ after 4 weeks of surgery). Liver steatosis developed in BKS-db/db mice and was significantly improved by RYGB procedure at 12 weeks, compared to the untreated and SHAM-PF liver (Fig 2c).

## RYGB improves β-cell viability, enhances insulin secretion and inhibits β-cell apoptosis

Plasma insulin was decreased in adult diabetic BKS-db/db mice (2.3 ± 0.5 ng/ml) as compared to young prediabetic BKS-db/db (8.1 ± 0.7 ng/ml) mice (Fig 3a). SHAM-PF fails to significantly rise plasma insulin levels (3.7 ± 0.35 ng/ml), and RYGB maintained plasma insulin levels

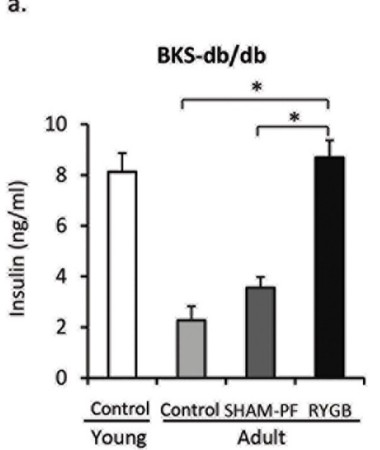
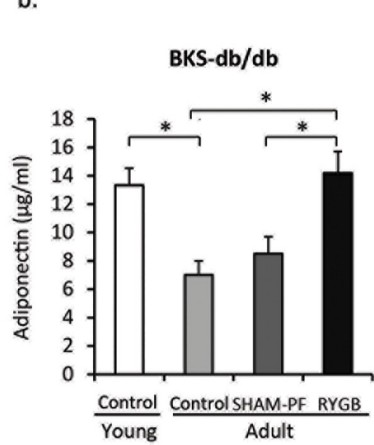

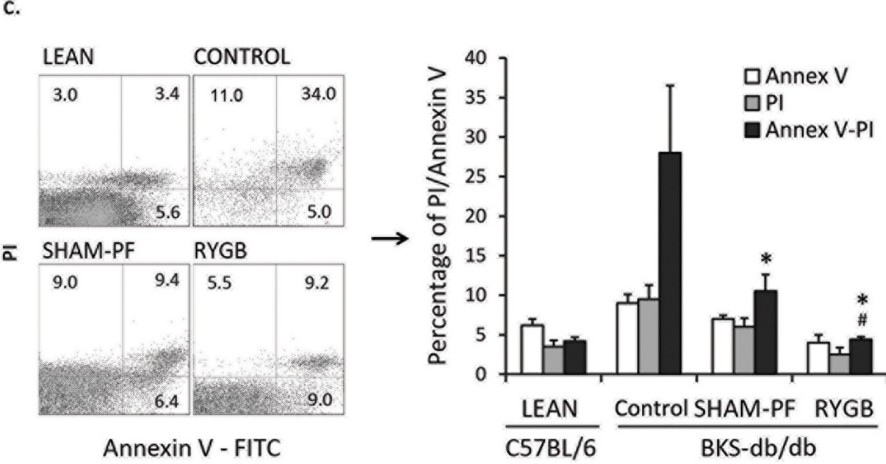

**Fig 3. RYGB maintains circulating insulin and adiponectin levels, preserves β-cell mass and prevents β-cell apoptosis when performed in young prediabetic BKS-db/db mice.** a. Circulating insulin levels. Young BKS-db/db mice (6 weeks old, n = 5); Adult BKS-db/db mice (12 weeks old, n = 5); SHAM-PF, sham surgery and pair-feeding (n = 5); and RYGB, BKS-db/db mice with RYGB at 6 weeks of age (N = 8). *: RYGB vs. SHAM-PF and Adult Control, $p < 0.01$. b. Circulating adiponectin levels. *: Young BKS-db/db vs. Adult BKS-db/db, and RYGB vs. SHAM-PF and Adult Control, $p < 0.01$. c. Annexin V flow cytometry analysis of apoptotic β-cells. LEAN, C57BL/6 mice; CONTROL adult BKS-db/db mice; SHAM-PF, BKS-db/db mice with sham surgery and pair-feeding, and RYGB, BKS-db/db mice with RYGB 4 weeks after surgery (n = 4 in each group). PI, propidium iodide positive; Annex-PI, Annexin V and PI double positive islet cells. *: SHAM-PF and RYGB vs. Control, $p < 0.001$; and [#]: RYGB vs. SHAM-PF, $p < 0.01$.

at 8.3 ± 0.3 ng/ml at 12 weeks post-surgery comparable to that in young prediabetic BKS-db/db mice (BKS-RYGB vs. BKS-Adult and BKS-SHAM-PF, $p < 0.001$, Fig 3a).

Adiponectin in the circulation, an anti-inflammatory adipokine, was reduced in adult BKS-db/db mice versus young BKS-db/db mice (7.0 ± 1.0 µg/ml vs. 13.3 ± 1.2 µg/ml, $p = 0.015$). RYGB maintained adiponectin at levels well-matched to young BKS-db/db mice (RYGB vs. Control and SHAM-PF, $p < 0.01$, Fig 3b).

To test whether RYGB improved β-cell apoptosis in BKS-db/db mice, islet cells were collected at 4 weeks post-surgery for Annexin V analysis by flow cytometry [28,32]. Increased apoptotic islet cells were found in control BKS-db/db mice (CONTROL, Fig 3d) compared to lean C57BL/6 mice (LEAN). Both SHAM-PF and RYGB reduced Annex V- and PI-double positive cells (Annex V-PI, $p <0.001$); however, when compared to SHAM-PF, RYGB further reduced apoptotic cells ($p < 0.01$, Fig 3d). The results show that RYGB improves β-cell viability and prevents β-cell apoptosis. These changes were also reflected by the observed changes in blood glucose and plasma insulin levels (Fig 3a).

## Gastric bypass improves β-cell mass and prevents diabetes in HFD-fed NZO mice

NZO male mice were fed chow in the first 6 weeks of age followed by HFD feeding to the end point (12 weeks). HFD accelerated body weight gain compared to chow (Fig 4a). Four of eight chow-fed and all HFD-fed NZO males developed hyperglycemia (fasting blood glucoses of $> 300$ mg/dl) after 6 weeks on HFD. SHAM-PF delayed but did not completely prevent hyperglycemia in 6 of 8 mice. RYGB prevented hyperglycemia in 100% of HFD-fed NZO mice

a.

b.

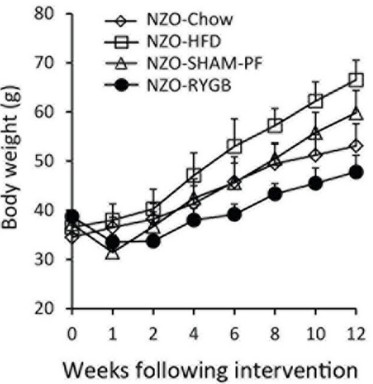
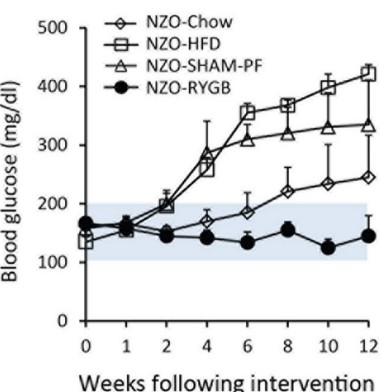

**Fig 4. RYGB prevents diabetes in NONcNZO10/LtJ (NZO) male mice.** a. RYGB resulted in weight loss in high-fat diet (HFD)-fed NZO mice. NZO-Chow, NZO chow-fed mice (n = 8); NZO-HFD, NZO mice with HFD starting at 6 weeks old (n = 8); NZO-SHAM-PF, sham surgery and pair feed with HFD starting at 6 weeks old (n = 8), and NZO-RYGB, RYGB with HFD starting at 6 weeks old (n = 10). NZO-RYGB vs. NZO-HFD, $p < 0.01$ from 2 weeks to 12 weeks; NZO-RYGB vs. NZO-SHAM-PF, $p < 0.05$ from 6 weeks to 12 weeks, and NZO-SHAM-PF vs. NZO-HFD, $p < 0.01$ from 4 weeks to 12 weeks post-surgery. b. RYGB improved hyperglycemia in HFD-fed NZO mice. NZO-Chow, chow-fed NZO mice (n = 8); NZO-HFD, HFD-fed NZO mice (n = 8); NZO-SHAM-PF, NZO mice with sham surgery and pair feeding (n = 8), and NZO-RYGB, HFD-fed NZO mice with RYGB (n = 10). NZO-RYGB vs. NZO-HFD and NZO-SHAM-PF, $p < 0.01$ from 6 weeks to 12 weeks post-surgery. The shaded area represents a normal range of lean mouse blood glucose levels.

(Fig 4b), suggesting that HFD accelerates the development of overt diabetes, which can be prevented by RYGB, and RYGB, but not SHAM-PF, preserves insulin[+] cells, contributing to improved hyperglycemia in HFD-fed NZO mice.

## RYGB improves glucose tolerance in C57BL/6-db/db null mice

C57BL/6 db/db nude mice (B6-db/db) have persistent insulin resistance without β-cell depletion [34,35]. To test whether RYGB improves insulin resistance in B6-db/db mice, RYGB was performed in adult B6-db/db mice ($\geq$ 12 weeks old), and IPGTT was performed starting at one-week post-surgery up to 12 weeks after surgery. Fig 5a shows that RYGB maintained normoglycemia through 12 weeks post-surgery (RYGB vs. B6-db/db-CONTROL and B6-db/db-SHAM-PF, $p < 0.05$). Fig 5b revealed that RYGB improved glucose tolerance through 12 weeks post-surgery (B6-db/db RYGB vs. B6-db/db-Control and B6-db/db-SHAM-PF, $p < 0.001$).

## Gastric bypass fails to improve glucose tolerance in genetic and chemically induced diabetic mice

To determine whether RYGB reverses diabetes when β-cells were severely damaged or exhausted, RYGB surgeries were performed in adult diabetic BKS-db/db mice ($\geq$ 12 weeks of age) with blood glucoses of $\geq$ 400 mg/dl. In contrast to young prediabetic BKS-db/db mice, RYGB did not improve hyperglycemia in adult diabetic BKS-db/db mice (Fig 5c).

STZ induced diabetes (blood glucoses of $\geq$ 300 mg/dl) in 75–80% of *MIP-luc* mice after 3–7 days of injection (referred to as day 0 in Fig 5d and 5e). Fig 5d indicates that RYGB failed to reverse hyperglycemia, and diabetic mice remained hyperglycemic through 4 weeks post-surgery. Bioluminescence imaging (Fig 5e) reveals reduced luciferase activity in STZ-induced diabetic mice, indicating decrease of β-cell activity, which cannot be improved by RYGB.

## Discussion

In prediabetes metabolic states such as obesity and glucose intolerance, β-cells biosynthesize and secrete more insulin in response to increased blood glucose level. With chronic hyperglycemia, there is a perpetually high demand for insulin, and β-cells eventually become exhausted, resulting in insufficient insulin synthesis and secretion causing overt diabetes [36]. The current study shows that RYGB preserves β-cell mass and prevents or slows the progression toward the insulin-deficient state, thereby preventing the development of overt diabetes in genetically modified prediabetic mice. However, RYGB neither reverses hyperglycemia nor improves β-cell viability in genetically modified diabetic mice and chemically induced diabetic mice. Gastric bypass must be performed before β-cells are completely exhausted, and it is ineffective in reversing the overt diabetic state once β-cells have already been severely damaged.

Although weight loss is associated with decreased insulin resistance, recent clinical results suggest that gastric bypass normalizes insulin sensitivity well before a significant reduction in body weight [37,38]. We have recently shown that RYGB induces sustained weight loss, fat mass reduction and improves insulin resistance in HFD-induced obese mice [16,17]. However, in BKS-db/db mice, body weight and fat mass are restored to pre-surgery levels from the second week post-surgery, and there is no statistical difference with SHAM-PF mice. However, while RYGB mice maintain normoglycemia post-surgery, SHAM-PF mice maintain hyperglycemia through the end point with a severe loss of β-cell mass, indicating that RYGB prevents the progression to overt diabetes and that this remarkable effect is independent of weight loss.

Consistent with a previous report that plasma insulin is decreased by 40% and 80% at 8 and 12 weeks of age, respectively [39], our findings show that plasma insulin in adult diabetic BKS-

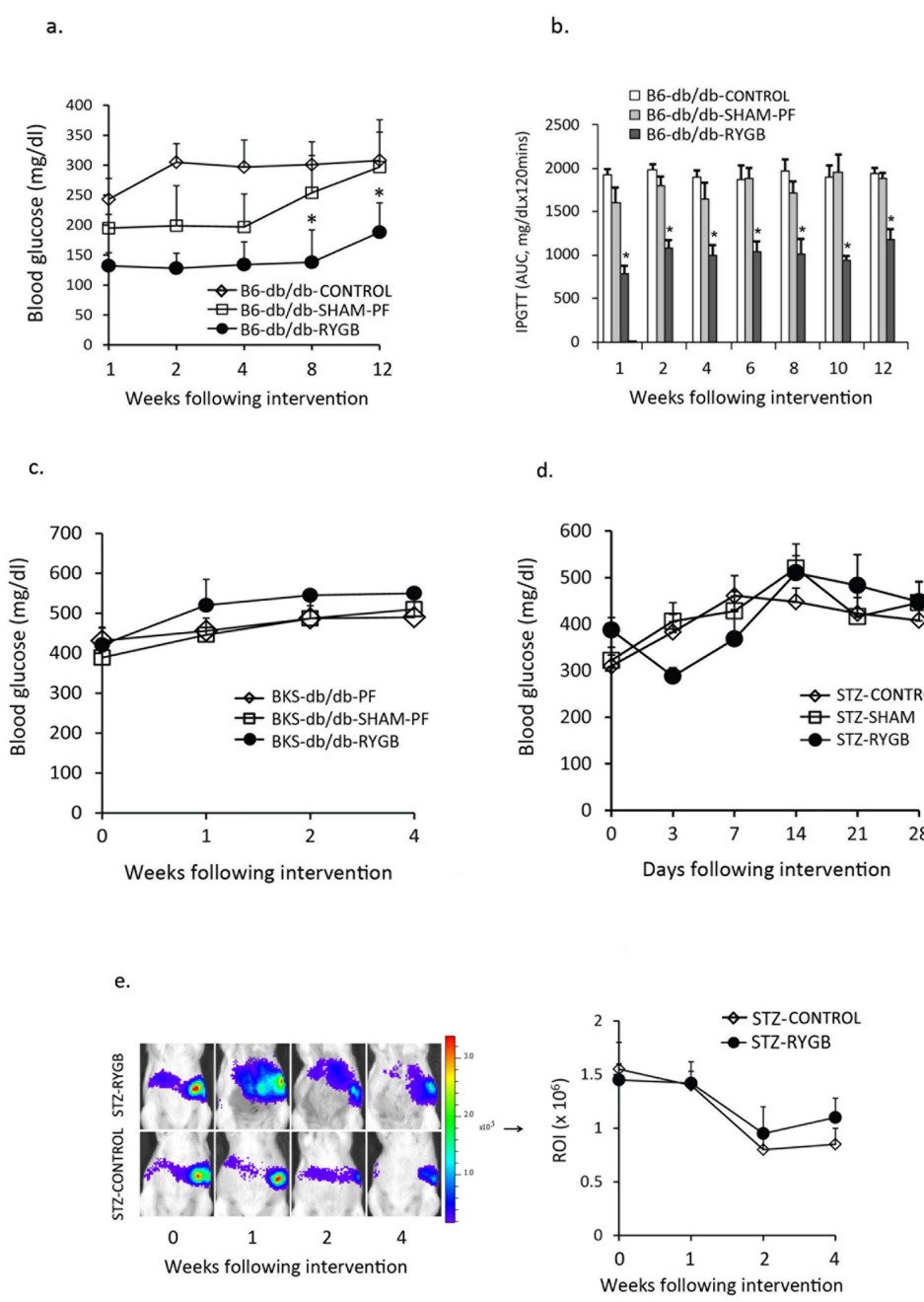

**Fig 5. RYGB improves hyperglycemia in adult B6-db/db mice, but not in adult diabetic BKS-db/db mice and chemically induced diabetic mice.** a. RYGB maintained normoglycemia in adult B6-db/db mice. *: B6-db/db-RYGB (n = 8) vs. B6-db/db-Control (n = 5) and B6-db/db-SHAM-PF (n = 5), $p < 0.05$. b. RYGB improved glucose tolerance (intraperitoneal glucose tolerance test) in adult B6-db/db mice. *: B6-db/db RYGB (n = 8) vs. B6-db/db-Control (n = 5) and B6-db/db SHAM-PF (n = 5), $p < 0.01$. c. RYGB failed to reverse hyperglycemia in adult diabetic BKS-db/db (12 weeks old) mice. BKS-PF, adult BKS-db/db mice with pair-feeding; BKS-SHAM-PF, adult BKS-db/db mice with sham surgery and pair-feeding, and BKS-RYGB, BKS-db/db mice with RYGB (n = 4 in each group). d. RYGB failed to reverse hyperglycemia in streptozotocin (STZ)-induced diabetic *MIP-luc* mice. STZ-Control, untreated STZ-induced diabetic mice (n = 5); STZ-SHAM, STZ-induced diabetic mice with sham surgery (n = 5), and STZ-RYGB, STZ-induced diabetic mice with RYGB (n = 6). e. Bioluminescence imaging (BLI) in STZ-induced diabetic *MIP-luc* mice. BLI was performed in *MIP-luc* mice one day before RYGB and at 1, 2 and 4 weeks after surgery. Luciferase is expressed as the regions of interest (ROI). STZ-Control, untreated STZ-induced diabetic mice (n = 6); and STZ-RYGB, STZ-indued diabetic mice with RYGB (n = 4).

db/db mice (12 weeks of age) is reduced to 25% of that in young prediabetic BKS-db/db mice (6 weeks of age). At the meantime, plasma adiponectin is also reduced to 50%. Adiponectin has been shown to induce ERK and Akt phosphorylation, stimulate insulin secretion and protect β-cells against apoptosis [40]. Gastric bypass preserves adiponectin expression, contributing to the prevention of β-cell exhaustion, and maintains normal insulinemia and normoglycemia when RYGB is performed in prediabetic BKS-db/db mice.

Both B6-db/db and BKS-db/db mice reveal increased blood glucose; however, B6-db/db mice glucose levels peak then decrease with aging due to β-cell hypertrophy and do not develop the full phenotype of T2D [15]. Gastric bypass improved glucose tolerance throughout the 12-week period in B6-db/db mice. T2D is characterized by a combination of genetic and epigenetic factors, and it is increasingly attributed to environmental factors. NONcNZO10/LtJ (NZO) is a new polygenic strain developed to more realistically model human obesity induced T2D. Consistent with previous reports [25,26], our results show that NZO male mice develop hyperglycemia and exhibited moderate to severe liver steatosis and islet atrophy in 40% of mice. HFD (60% fat)-feeding results in hyperglycemia in 100% of NZO male mice. Insulin$^+$ cells are reduced in HFD-induced diabetic NZO mice due to β-cell apoptosis, as we found in the BKS-db/db model. RYGB preserves β-cell mass, maintains normoglycemia and prevents HFD-induced diabetes, suggesting that genetic- and HFD-mediated diabetes can be prevented by gastric bypass.

STZ targets insulin-producing β-cells in the pancreas and results in severe damage of β-cells in mice, mimicking T1D phenotypes. Recent studies have suggested that gastric bypass yields metabolic benefits as well as leads to better cardiovascular outcomes for the treatment of T1D [41,42]. However, our results show that gastric bypass neither rescues damaged β-cells nor improves hyperglycemia in STZ-induced diabetic mice. We understand that one major limitation of our findings relates to the different proliferation of β-cells, *i.e.* many mitogens, growth factors and nutrients have been shown to induce expansion in rodent models. However, human β-cells show poor responsiveness to these stimuli [43]. The definition of prediabetes in the clinic is that blood glucose level (100 to 125 mg/dL) is higher than it should be (in normal condition) but not high enough to diagnose diabetes ($\geq$ 126 mg/dL) [44]. Consequently, the gastric bypass related results in mice must be carefully interpreted concerning human implications.

In conclusion, gastric bypass results in sustained normoglycemia and improved glucose tolerance associated with preservation of β-cells in prediabetic BKS-db/db mice. These effects are independent of weight loss. Gastric bypass also prevents the development of overt diabetes in HFD-fed NZO mice. In adult diabetic BKS-db/db mice, however, gastric bypass neither reverses diabetes nor improves β-cell viability. Similarly, gastric bypass did not reverse STZ-induced overt diabetes. Our findings suggest that RYGB improves hyperglycemia and must be performed before β-cells are severely damaged.

## Supporting information

**S1 File.**
(DOCX)

## Author Contributions

**Formal analysis:** Deng Ping Yin.

**Investigation:** Chenyu Zhu, Rui Xu, Yuxin Li, Michael Andrade, Deng Ping Yin.

**Methodology:** Deng Ping Yin.

**Supervision:** Deng Ping Yin.

**Validation:** Deng Ping Yin.

**Visualization:** Deng Ping Yin.

**Writing – original draft:** Deng Ping Yin.

**Writing – review & editing:** Michael Andrade, Deng Ping Yin.

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
