## [Decision Letter · Decision Letter 0]

7 Jul 2021

PONE-D-21-17693

Gastric Bypass Prevents Diabetes in Genetically Modified Mice and Chemically Induced Diabetic Mice

PLOS ONE

Dear Dr. Yin,

Thank you for submitting your manuscript to PLOS ONE. After careful consideration, we feel that it has merit but does not fully meet PLOS ONE’s publication criteria as it currently stands. Therefore, we invite you to submit a revised version of the manuscript that addresses the points raised during the review process.

We look forward to receiving your revised manuscript.

Kind regards,

Michael Bader

Academic Editor

PLOS ONE

Journal Requirements:

“This work was in part supported by the Department of Surgery at University of Chicago to the 328 Animal Microsurgery Center and by the National Institution of Health (NIH) grant DK020595 to 329 the Metabolic Core of the University of Chicago and the Tissue and Cell Models Core of the 330 University of Chicago DDRCC (P30 DK42086).”

 “NIHDDK DK020595 to the Metabolic Core of the University of Chicago

NIDDK P30 DK42086 to the Tissue and Cell Models Core of the University of Chicago DDRCC.”

5. We note that Figure 3C and 4C in your submission contain map images which may be copyrighted. All PLOS content is published under the Creative Commons Attribution License (CC BY 4.0), which means that the manuscript, images, and Supporting Information files will be freely available online, and any third party is permitted to access, download, copy, distribute, and use these materials in any way, even commercially, with proper attribution. For these reasons, we cannot publish previously copyrighted maps or satellite images created using proprietary data, such as Google software (Google Maps, Street View, and Earth). For more information, see our copyright guidelines: http://journals.plos.org/plosone/s/licenses-and-copyright.

a. You may seek permission from the original copyright holder of Figure 3C and 4C to publish the content specifically under the CC BY 4.0 license. 

Reviewers' comments:

Reviewer's Responses to Questions

**Comments to the Author**

1. Is the manuscript technically sound, and do the data support the conclusions?

Reviewer #1: Yes

Reviewer #2: No

2. Has the statistical analysis been performed appropriately and rigorously? 

Reviewer #1: Yes

Reviewer #2: Yes

3. Have the authors made all data underlying the findings in their manuscript fully available?

Reviewer #1: Yes

Reviewer #2: Yes

4. Is the manuscript presented in an intelligible fashion and written in standard English?

Reviewer #1: Yes

Reviewer #2: Yes

5. Review Comments to the Author

Reviewer #1: In this study, Zhu et al use different diabetic/pre-diabetic mouse models to determine at what stage Roux-en-Y gastric bypass surgery restores pancreatic beta cell function/survival. They found in young (6 week-old) BKS db/db mice that RYGB lowered FBG, improved IPGTT and ameliorated fatty liver in a food intake-independent manner. This was associated with a normalization of insulin release and expression and reduced pancreatic beta cell death. Similar findings were made in DIO mice and adult (12 week) B6 db/db mice but not in adult BKS db/db mice or in mice with chemical ablation of pancreatic beta cells from STZ Treatment. It is concluded that RYGB should be performed as early as possible in prediabetic individuals before the exhaustion of pancreatic beta cells for an optimal outcome.

This is an excellent study from an estalished group in mouse bariatric surgery that is well reasoned, controlled and executed. The main text and results are easy to follow and the findings are thorough, convincing and robust. The authors can improve their manuscript by speculating in the discussion section which factors could be responsible for the weight loss-independent improvment in pancreatic beta cell function/survival. While this has not been extensively studied for RYGB in mice, it seems that GLP-1 is not essential for VSG (PMID 29759973). There is also the study showing weight loss-independent improvement pancreatic beta cell function after VSG which should be discussed in the light of their findings (PMID 30777938). Also, in Table 1 it says db/db mice (BKS and B6) were only operated on at 6 weeks of age. However in Figure 5 These mice had surgery at 12 weeks of age. Please clarifiy.

Reviewer #2: About the introduction

The aim of the paper of Zhu C and Zhu R et al is to study the capacity of gastric bypass to prevent diabetes in various pre-diabetic rodent models. The paper is original but very descriptive and there are many concerns about the conclusion.

-concerning mice rodent models :

-line 67: “An established mouse RYGB model induces significant weight loss, reduces body fat, improves insulin sensitivity and ameliorates liver steatosis in high fat diet induced obese mice”. It is surprising that in this procedure, there is no increase of insulin secretion. Indeed, the characteristic of gastric bypass surgery is to restore beta cell function quickly after the procedure and before any significant body weight loss. This has been established by Walter Pories in humans (1995) and replicated in various rodent models. Thus, the results of the present paper dedicated to the study of beta cell function after a surgery can be challenged if we consider that the procedure has no demonstrated effect on insulin secretion. How the authors may explain this difference of their procedure when compared to the literature ?

-line 72: the db/db mice is cited as an important model with various stages of diabetes. But, ob/ob mice is also interesting with similar characteristics. Importantly, bariatric surgery has been performed by different teams in ob/ob mice and these papers showed that beta cell function recovery may be possible despite lack of effect of surgery on body weight. In addition, beta cells of ob/ob mice activated molecular pathways able to reduce the deleterious effects of oxidative stress while db/db mice did not display such mechanisms. This explain why mean glucose levels are higher in db/db mice when compared to ob/ob mice. Some references about ob/ob mice and bariatric surgery may be added.

-line 89-90: ….(STZ), mimicking tye 1 diabetes (T1D) phenotype.

STZ is a toxic compound for beta cells. Consequently, STZ induces a model of deficit of insulin secretion but not associated with autoimmunity, the specific feature of T1D. Even if a local inflammation in pancreatic islets can be observed the few days following STZ administration, such inflammation is different that of insulitis of T1D and is not driven by the same immune cells. Thus, STZ rodent model is a model of diabetes lacking insulin without auto-immunity. This is important if we consider the potential proliferation of new beta cells from ductal cells after STZ administration and bariatric surgery, mechanism not possible when auto-immunity is present. This may change insulin secretion capacity.

Results

-line 177: the figure 1B show that decrease of body weight is similar in RYGB and SHAM-PF db/db mice. I don’t understand line 179 the p< 0.05.

-line 180 figure 1c: db/db mice have higher amount of fat mass than lean. During the follow-up, there is no difference between fat mass in Sham-PF and RYBG groups. Is it correct ?

-line 186: glucose tolerance n RYGB is improved in comparison to control after the first tow weeks of follow-up. Th figure 2B showed a progressive increase of AUC glucose in RYGB mice suggesting a progressive deterioration of glucose homeostasis. This has to be discussed

-line 183: an absolute level of insulin without the glucose levels cannot be correctly analyzed. It is mandatory to plot glucose and insulin levels (or AUC glucose). There are no details about the metabolic status of mice (fasted or random fed).

-line 19: figure 3B : on this figure, the significant difference between young controls and adult controls has to be added.

-line 205: figure 3c showed representative pancreata images. This is not sufficient and quantification of insulin +,cells has to be showed and expressed by the weight of pancreas (because it cannot be excluded that surgery may modify the total weight of pancreas by changes in exocrine function). It is the same for figure 4c.

-figure 4 and figure 5: all the figures with glucose levels during the follow-up or the IPGTT (figure 5b) needed the insulin secretion levels. It is difficult to conclude if the results of immunohistochemistry staining or those of bioluminescence imaging may have in vivo consequences on glucose levels. Extrapolation of in vivo insulin secretion from immunostaining is impossible and such extrapolation deserves the paper.

Thus, in vivo insulin secretion has to be studied and showed in this paper. In addition, the SOS study has showed in humans that gastric bypass may reduce incidence of T2D in non diabetic obese patients during a long-term follow-up suggesting that the prevention of diabetes is possible in humans. Thus, the present paper does not add novelty in this field.

6. PLOS authors have the option to publish the peer review history of their article (what does this mean?). If published, this will include your full peer review and any attached files.

Reviewer #1: No

Reviewer #2: No

---

## [Editor Report · Decision Letter 1]

11 Oct 2021

Gastric bypass prevents diabetes in genetically modified mice and chemically induced diabetic mice

PONE-D-21-17693R1

Dear Dr. Yin,

We’re pleased to inform you that your manuscript has been judged scientifically suitable for publication and will be formally accepted for publication once it meets all outstanding technical requirements.

Kind regards,

Michael Bader

Academic Editor

PLOS ONE
---

## [Editor Report · Acceptance letter]

13 Oct 2021

PONE-D-21-17693R1 

Gastric bypass prevents diabetes in genetically modified mice and chemically induced diabetic mice 

Dear Dr. Yin:

I'm pleased to inform you that your manuscript has been deemed suitable for publication in PLOS ONE. Congratulations! Your manuscript is now with our production department. 

Kind regards, 

on behalf of

Prof. Michael Bader 

Academic Editor

PLOS ONE